# Unsupervised Deep Comparison For Face Anomaly Appraisal

Abdullah Hayajneh[1], Erchin Serpedin[1], and Mitchell A. Stotland[2]

*Abstract*—This paper proposes an innovative anomaly appraisal framework that combines machine learning and signal processing for the consistent detection, localization, and evaluation of anomalies in human faces. The primary objective of this framework is to create a universal and objective metric for evaluating the degree of facial anomaly and reconstructive surgical outcomes. This metric should be well aligned with the human assessments. To accomplish this, the framework leverages the StyleGAN2 facial generator to normalize human faces that may exhibit diverse deformities. The proposed method utilizes a pre-trained Convolutional Neural Network (CNN) to extract and compare deep anomalous features between the original image and its normalized counterpart, in an unweighted manner. The resulting anomaly maps are merged into a heatmap, effectively highlighting the abnormal facial regions. This heatmap is then employed to generate a machine score, quantifying the degree of anomaly in the face. In order to assess the effectiveness of the proposed method, a comprehensive comparison of the generated anomaly maps is conducted using metrics such as Learned Perceptual Image Patch Similarity (LPIPS), Structural Similarity Index (SSIM), Pixelwise Subtraction (PS), alongside the newly introduced framework. The conducted comparative analysis demonstrates the framework's robust performance, showing a high linear correlation (0.92 Pearson's r score) and a strong monotonic relationship (0.85 Spearman's $\rho$ score) between the human and machine generated scores. These results corroborate the framework's efficacy and its close alignment with the human judgment in assessing different levels of facial anomalies.

## I. Introduction

### A. Clinical Relevance of Facial Anomaly Detection

*Background:* Facial clefts and non-cleft congenital facial anomalies occur in approximately 1 in 500 births [4], [30], [35], while many more children experience facial deformities due to post-traumatic or post-oncologic causes. These conditions are in general associated with difficulties in breathing, feeding, and speech, and present a significant psychosocial burden [8], [15].

*Problem Definition:* To improve their condition, patients with facial difference seek unbiased evaluations and corrective surgical interventions. During the pre-operative assessment, reconstructive surgeons must be able to identify and objectively measure the extent of a patient's facial anomaly relative to what would be considered their unaffected face. Then, in the post-operative period surgical outcomes need to be fairly evaluated. Currently, however, healthcare providers, patients, and third-party payers are unable to objectively determine the effectiveness of treatment because no universally accepted framework for such assessment exists. A variety of facial measurement techniques have been described over the years (e.g., anthropometric landmark measurements, crowd-sourced surveys, expert ratings, patient-reported outcome measures, and eye tracking studies [5], [7], [10], [11], [16]–[18], [22], [25], [26], [28], [31]–[33], [40]). All of these methods suffer from any combination of subjective judgement, technology-dependency, time inefficiency, cost, or unsuitability for real-time clinical use.

*Impact:* This paper describes an AI-powered model that can objectively and reproducibly appraise a human face in terms of form, deformity, and the impact of surgical alteration. We believe this has the potential to become a long sought after universal clinical tool allowing for fair comparison between the outcomes of facial reconstructive techniques and surgeons; opening up new avenues for discussion between providers and patients; and helping to justify the value of facial reconstructive surgery to healthcare payers.

### B. Modern Facial Anomaly Detection Approaches

Various computational approaches have been developed for identifying and localizing anomalies [9], [23], [29], [34], [41]. However, these methods often face challenges such as increased complexity, difficulties handling high-dimensional datasets with limited samples, and reliance on strong supervision. In addition, generating binary ground truth for facial anomaly detection is difficult. Nevertheless, methods have been proposed for forecasting anomalous regions in images using public datasets like MVTecAD [3], BTAD [27], AITEX [39], BrainMRI [36], and HeadCT [36].

In general, anomaly detection can be construed as the process of gauging the similarity between a normal reference and an aberrant counterpart [37], [38]. The creation of a reliable ground truth anomaly map for the human face, however, can be particularly problematic because the age, gender, race, and type of deformity captured in a facial image can evoke strong emotional responses in the observer. Moreover, geographically smaller anomalies can have a greater impact on appearance

---

[1]*ECEN Dept., Texas A&M University, College Station, TX, 77843.*
[2]*Surgery Dept., Sidra Medicine, Doha, Qatar.*

than larger ones, depending on their anatomic location and salience.

Our group has developed contemporary facial anomaly detection approaches that find the closest representation of an original image within the latent space of a generative model (i.e., StyleGAN2 generator [20]). The assessment of disparity between the original and normalized samples has been conducted by determining the pixelwise subtraction error (PSE) or via other similarity detection methods [6], [13]. Direct and indirect approaches have been proposed to quantify the degree of facial anomaly. Indirect methods employ regression models to transform extracted features from the "raw" and "normalized" versions of an image into an index characterizing the divergence between the two images [6]. Direct approaches connect the pixelwise residual image to a numerical value using simple distance measures, such as the $l_2$ norm [13], [37]. Direct approaches offer interpretability by directly linking the machine scores with the highlighted anomalous regions in the heatmap.

Despite a strong correlation with collected human ratings of the images, the technique outlined in [13] yields a residual image dependent solely on shallow pixelwise color differences between the original facial image and its normalized version. Consequently, for any small color intensity difference between two blobs of pixels in the original and normalized sample, the pixelwise subtraction signal may be too small and indistinguishable from other image textures. For example, the skin tone and color of the nose as well as the cutaneous and vermilion portions of the upper lip are often quite similar in their color intensity values. A pixelwise subtraction method, therefore, may not be sensitive enough to accurately localize and measure subtle – though clinically important – cleft lip deformities, for example. Moreover, pixelwise comparisons consider differences only within infinitesimal regions of an image and fail to capture relationships between different regions (see Fig. 1). In addition, the method proposed in [13] assumes post-processing of the heatmaps to enhance them and reduce the noise, potentially leading to performance losses. To address this issue, an alternative technique that does not rely upon pixel color information and is capable of integrating higher-level features for multiscale comparison is essential.

## II. RELATED WORK

To create an anomaly detection method that is fit for clinical applications and is independent of color information, we explored a range of techniques. Shallow approaches such as Structural Symmetry Index (SSIM) [42] and Peak Signal-to-Noise Ratio (PSNR) [14], as well as deep approaches such as the Learned Perceptual Image Patch Similarity (LPIPS) metric [43] have been implemented as similarity measures. However, these methods may not offer a clear separation between normal

and abnormal pixel information when comparing two related facial images.

The fundamental principle behind deep feature comparison involves feeding two images into a convolutional neural network (CNN, e.g., AlexNet) to extract deep feature maps that collectively aid in the object detection. LPIPS applies a learned weighted sum of the difference of these deep feature maps to enable image comparison. However, LPIPS may not identify all abnormal differences because the learned weights are specialized to detect specific object differences.

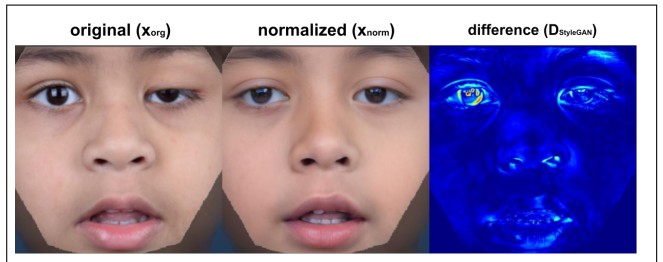

Fig. 1. PSE does not highlight low-varying differences between the original and the normalized image. The IRB approval-written informed consent-was granted to report this case details.

### A. Current Work

To overcome these limitations, a novel anomaly appraisal framework employing deep feature comparison is proposed herein. The proposed method initially normalizes an input image through an optimization process conducted within the latent space of a pre-trained StyleGAN2 facial generator. Subsequently, a pre-trained Convolutional Neural Network (CNN) is employed to compare the original image with its normalized version. During the comparison process, deep anomalous features between the two images are extracted and merged into a comprehensive heatmap without weighting. A machine score is then computed to quantify the degree of similarity between the images. For this approach, there is no need to train a linear transformation model to account for the importance of the extracted feature maps, as opposed to the LPIPS framework. This approach employs fewer processing steps which in turn increase the model's robustness to noise.

It is shown that the proposed framework detects anomalous regions of the face more acutely and in a manner correlating better with human judgment. To illustrate the framework's potential as a universal noise extractor from a wide range of images, we investigate in depth its capability to identify and delineate cleft lip and other types of congenital facial deformities.

In summary, this work proposes the following key contributions:

- A simple but effective image comparison architecture to enable the detection and localization of anomalies in human faces.

- A new anomaly appraisal framework that uses the unweighted deep difference maps to generate image normality scores, without any post-processing of the maps.
- A demonstration of the effectiveness of directly using the deep feature maps to compare images, without training a model to weigh the deep features according to their importance.
- New insights into the effectiveness and limitations of different similarity measures in detecting differences between two images.

## III. MATERIALS AND METHODS

This study was approved by the Institutional Review Boards (IRBs) of Sidra Medicine ($1855398, 1830649$) and Texas A&M University ($IRB2023-0111, MODCR00000073$). Let $x_{org} \in \mathbb{N}^{n \times m \times c}$ denote the original image of a human face with a cleft lip deformity, with $n$, $m$ and $c$ representing its height, width, and the number of color channels, respectively. The research objective is to obtain a heatmap $D \in \mathbb{R}^{n \times m}$ highlighting the anomalous information contained in $x_{org}$. To obtain the heatmap $D$, a complimentary normalized image $x_{norm}$ is first generated by applying off-the-shelf image latent inversion methods on $x_{org}$ [1], [2], [19], [24].

The overall framework for outlier detection, localization, and measurement is illustrated in Fig. 2b. This framework adapts the model previously proposed in [13] by exchanging the shallow pixelwise subtraction method for a deep feature map processing. The workflow begins with a CNN model pretrained on a dataset of human faces and follows these steps.

### A. Image Preprocessing

This stage encompasses essential activities such as scaling, translation, orientation correction, cropping, and color adjustments. These activities collectively prepare the original image ($x_{org}$) for the subsequent normalization step.

### B. Image Normalization

Normalized image $x_{norm}$ is attained using the Style-GAN2 Projection Algorithm. The normalized facial image is then refined using the Model Adaptation Algorithm outlined by the following two sub-steps.

*1) Latent Space Projection:* The original facial image $x_{org}$ is normalized by projecting it on the latent space of the StyleGAN2 facial generator $G$ [20]. This process is carried out by first searching in the latent space of the StyleGAN2 generator the latent vector $w \in \mathbb{R}^{512}$ that produces the closest face $x_{norm}$ to $x_{org}$. The algorithm ensures that $w$ leads to the generation of $x_{norm}$ and that for each noise map $\mathbf{n_i}$ in $G$ its entries are uncorrelated [21]. This implies that $x_{norm}$ does not depend on the noise maps that are sampled from a uniform distribution in the StyleGAN2 model. A summary of the code is provided in Algorithm 1.

*2) Pretrained Model Adaptation:* After obtaining the initial latent vector $w$, additional training is carried out to adapt the parameters of the generator $G$ to reconstruct additional identity-preserving details in $x_{org}$ [13]. This operation is conducted by freezing the latent vector $w$ and adapting $G$. The adaption of network $G$ relies on the composite loss function:

$$
\begin{aligned}
\mathcal{L}(x_{org}, x_{norm}) &= \mathcal{L}_{LPIPS}(x_{org}, x_{norm}) \\
&+ \mathcal{L}_{l_2}(x_{org}, x_{norm}).
\end{aligned}
$$

The model adaptation process is coded in Algorithm 2.

---

**Algorithm 1** Face Inversion, Input: Face image $x_{org}$ and StyleGAN2 generator model $G$. Output: Latent vector $w$ and set of noise maps $\mathbf{n} = \{\mathbf{n_1}, \ldots, \mathbf{n_m}\}$, $r_i = 2^i$, $i \in \{1, \ldots, m\}$, $m = 18$, $\alpha = 10^5$, $N = 10^4$.

---

$Z = \{Z_1, ..., Z_N\} \leftarrow U\{1, ..., N\}$
$W = \{W_1, ..., W_N\} \leftarrow G(Z)$
$w \leftarrow \frac{\sum_n (W_n)}{N}$
$\mathbf{n} = \{\mathbf{n_1}, ..., \mathbf{n_m}\} \leftarrow \{\mathbb{R}^{r_1 \times r_1}, ..., \mathbb{R}^{r_m \times r_m}\}$
**while** not converge **do**
    $\mathcal{L}_{Image} = \mathcal{D}_{LPIPS}(x_{org}, G(w, \mathbf{n}))$
    For $j = 1, \ldots, m$
      For $k = 1, \ldots, j - 1$
        $\mathbf{n}_j^{(k)} \leftarrow Downsample(\mathbf{n}_j, 0.5)$
        $r_{j,k} \leftarrow \frac{r_j}{2^k}$
        $\mathcal{L}_{j,k} \leftarrow (\frac{1}{r_{j,k}^2} \sum_{x,y} \mathbf{n}_j^{(k)}(x,y) \mathbf{n}_j^{(k)}(x-1,y))^2 +$
        $(\frac{1}{r_{j,k}^2} \sum_{x,y} \mathbf{n}_j^{(k)}(x,y) \mathbf{n}_j^{(k)}(x,y-1))^2$
      End For
    End For
    $\mathcal{L}(x_{org}, G(w, \mathbf{n})) = \mathcal{L}_{Image} + \alpha \sum_{j,k} \mathcal{L}_{j,k}$
    $\nabla w \leftarrow \frac{d\mathcal{L}}{dw}$
    For $j = 1, \ldots, m$
      $\nabla \mathbf{n}_j \leftarrow \frac{d\mathcal{L}}{d\mathbf{n}_j}$
      $\mathbf{n}_j = \mathbf{n}_j - \nabla \mathbf{n}_j$
    End For
    $w = w - \nabla w$
**end while**
**return** $w, \mathbf{n}_1, \mathbf{n}_2, ... \mathbf{n}_m$

---

### C. Extraction of Difference Maps

To effectively identify the anomalous differences between $x_{org}$ and $x_{norm}$, deep feature maps $\{x_l^k\}$ are derived by passing both images through two identical feature extractors $F_1, F_2 : \mathbb{N}^{n \times m \times c} \rightarrow \mathbb{R}^{\frac{n}{2^{(L-1)}} \times \frac{n}{2^{(L-1)}} \times k}$, with $k$ representing the number of feature maps in the final remaining layer.

All the extracted feature maps are then resized to match the size of $x_{org} \in \mathbb{N}^{n \times m}$, and the subtraction is performed to obtain the set of difference maps $\{D_l^{(k)}\}$ via the following equation:

$$
D_l^{(k)} = ((x_{org,l}^{(k)} - x_{norm,l}^{(k)}) \odot (x_{org,l}^{(k)} - x_{norm,l}^{(k)}))^{\circ 2}, \quad (1)
$$

**Algorithm 2** Pretrained model adaptation. Input: Face image $x_{org}$, its corresponding closest latent vector $z \in \mathbb{R}^{512}$ and a StyleGAN2 generator $G$. Output: Adapted generator $G'$.

---

$\quad G' \leftarrow G$
$\quad$**while** not converge **do**
$\quad\quad x_{norm} \leftarrow G'(z)$
$\quad\quad \mathcal{L}(x_{org}, x_{norm}) = \mathcal{L}_{LPIPS}(x_{org}, x_{norm}) +$
$\quad\quad\quad\quad\quad\quad\quad\quad \mathcal{L}_{l_2}(x_{org}, x_{norm})$
$\quad\quad \nabla g \leftarrow \frac{d\mathcal{L}}{dz}$
$\quad\quad G' = G' - \nabla g$
$\quad$**end while**
$\quad$**return** $G'; x_{norm}$

---

with $l$, $k$, $\odot$, and $\circ^2$ representing the layer index, filter index within the layer, elementwise (Hadamard) multiplication and squaring operation, respectively. For the Alexnet CNN, a total of 1458 difference maps are generated.

### D. Resizing the Difference Maps

Assume the output of each convolutional layer represents a set of feature maps $x$, whose size $\left(\frac{n}{2^l} \times \frac{m}{2^l}\right)$ keeps decreasing by a factor of 2 while increasing the layer index $l$. As for LPIPS CNN-based similarity measures, the proposed framework employs the upsampling operation on the feature maps for all the CNN layers except the first layer to remain consistent with the original image size, $x_{resized} \in \mathbb{R}^{n \times m}$. Each feature map is scaled with a suitable factor depending on the current feature map resolution in the convolutional layer. The process of upsampling allows the capturing of features at different scales while simultaneously providing an accurate representation of the noise in its original locations. Fig. 2a depicts a schematic representation of the proposed architecture.

### E. Overall Score Calculation

The overall difference map is defined as the average of all the difference maps:

$$D = \frac{1}{LK} \sum_{l=0}^{L} \sum_{k=0}^{K} D_l^{(k)}. \tag{2}$$

Given the overall difference map $D$, the score is calculated via this new metric:

$$s(D) = -log\left(\frac{1}{mn} \sum_{j=1}^{n} \sum_{i=1}^{m} D_{i,j}\right). \tag{3}$$

In (3), we consider the logarithm of the overall difference map to account for small and subtle facial differences that may be observed by humans. The metric is designed to be less sensitive to large valued differences and more sensitive to small and subtle changes. The negative sign is to ensure a positive measure of (Pearson) correlation.

TABLE I
INFORMATION ABOUT THE DATASETS USED TO EVALUATE THE PROPOSED
FRAMEWORK.

| Dataset 1 | #Samples | Original resolution | Source |
|---|---|---|---|
| Right cleft lips | 40 | | |
| Left cleft lips | 17 | $6016 \times 4016$ | Sidra Medicine |
| Bilateral cleft lips | 8 | | |
| Normal samples | 55 | $1024 \times 1024$ | StyleGAN2 |
| Total | 120 | | |
| Dataset 2 | | | |
| Micrognathia | 3 | | |
| Craniofrontonasal dysplasia | 2 | | |
| Congenital ptosis | 8 | | |
| Facial nevus | 3 | $6016 \times 4016$ | Sidra Medicine |
| Craniosynostosis | 7 | | |
| Facial palsy | 4 | | |
| Facial syndromes | 14 | | |
| Other facial anomalies | 14 | | |
| Normal samples | 55 | $1024 \times 1024$ | StyleGAN2 |
| Total | 110 | | |

## IV. EXPERIMENTS

Two datasets of faces were used to evaluate the proposed framework. The dataset 1 contains normal and cleft faces, while the dataset 2 consists of normal and other general facial anomalies. The anomalous images in the datasets were obtained from the practice of one co-author, with the approval of the Institutional Review Board and the signed informed consent of all patients/parents. Information about number of samples, image resolution and provider is shown in Table I. Both datasets were presented to a number of volunteers to get 20+ ratings for each sample. Volunteers have rated the appearance of the images from 1 (least normal) to 7 (most normal) in two separate surveys. The mean of the ratings for each image was calculated for the analysis.

To evaluate the ability of the proposed framework to localize and measure facial anomalies, and to establish the optimal CNN for the proposed framework, we ran a series of tests using the 125 clinical images in dataset 1, portraying various types of cleft lip deformity. The state-of-the-art AlexNet CNN and its LPIPS version were tested and compared. The fully connected layer at the end of each network was removed and used as a feature extractor. Heatmaps were generated, including a difference map between $x_{org}$ and $x_{norm}$, and a machine score was produced for each of the 125 facial images. The human and machine scores were compared using the Pearson correlation coefficient. Afterward, the proposed framework with the best-performing CNN was compared against the SSIM, PSE, and LPIPS-AlexNet without post-processing. Pearson and Spearman correlation coefficients were used to examine how machine and human scores are linearly and monotonically related, respectively. Also, we have included Patchcore [34], a popular and recent anomaly detection method for comparison. To match the generated scores with the Ground-Truth (GT) scale, the following normalization process is carried out: (i)

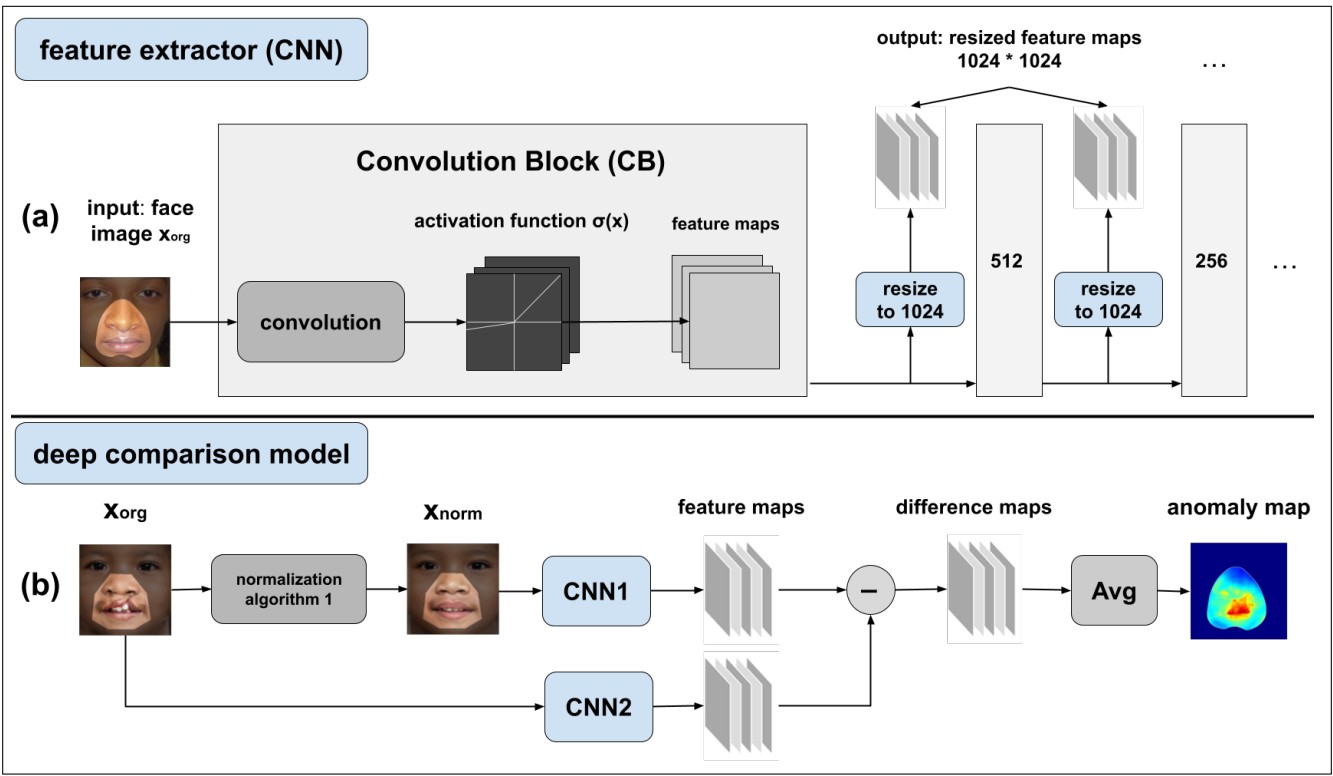

Fig. 2. (a) Feature extractor architecture. (b) Overall framework of the proposed anomaly detector.

Find three human faces with the highest GT (human) ratings. Calculate the average of their machine scores. (ii) Find the average of the three lowest GT ratings similarly. (iii) Use the roughly estimated range to normalize the rest of the scores. These steps allow to measure how aligned the normalized values are to the GT human ratings. For this purpose, the Mean Absolute Error (MAE) and the Wasserstein distance between the histograms of the machine/human scores were measured. All analysis was conducted on Python 3.6 using Pytorch, Opencv, Scipy, and Skimage libraries on Intel i7-10751H CPU 2.6 GHz with Nvidia GeForce 2080 Super with Max-Q design.

## V. RESULTS

Fig. 3a consists of three facial images depicting unique facial deformities (congenital melanocytic nevi in the two top images and micrognathia in the bottom image). It can be appreciated that the pigmented skin lesions and the hypoplastic lower jaw region are far more acutely highlighted when using the proposed framework than PSE and SSIM. Table II shows the correlations between the machine and human scores obtained by using the AlexNet CNN base model for heatmap generation, comparing its weighted maps version in LPIPS implementation versus the original AlexNet CNN. The proposed model demonstrated a clear advantage. The maximum machine:human image rating correlation was

obtained by using the original AlexNet CNN ($r = 0.92$) when the layer 4 feature maps were averaged. For all CNNs tested configurations, the correlation between machine and human image ratings was less tightly correlated when using the LPIPS models (e.g., for LPIPS-AlexNet the correlation dropped to 0.913).

TABLE II
THE PEARSON CORRELATION BETWEEN HUMAN AND MACHINE SCORES FOR 65 CLEFT LIP AND 55 NORMAL IMAGES SHOWS THE IMPACT OF USING THE WEIGHTED ALEXNET (LPIPS) VERSUS PROPOSED UNWEIGHTED ALEXNET MAPS. THE BEST COMBINATIONS ARE IN BOLD.

| layer | Oral/Nasal Region Only | | Entire Face | |
|---|---|---|---|---|
| | LPIPS-AlexNet | AlexNet | LPIPS-AlexNet | AlexNet |
| $l = 1$ | 90.4 | 91.3 | 85.7 | 87.5 |
| $l = 2$ | 82.9 | 87.4 | 75.5 | 82.3 |
| $l = 3$ | 90.4 | 91.6 | 85.6 | 87.6 |
| $l = 4$ | 91.3 | **92.0** | 87.3 | **87.8** |
| $l = 5$ | 89.3 | 90.6 | 86.3 | 86.7 |

Table III shows the Wasserstein distance ($W_1$) between the normality histogram of the human ratings and each of the histograms obtained from Patchcore, SSIM, LPIPS-AlexNet (all layers), PSE, and AlexNet (proposed). Also, the MAE, Spearman ($\rho$), and Pearson ($r$) correlation measures for these methods are reported. As can be observed, there is a notable advantage of the proposed method in all the performance metrics against the other similarity methods.

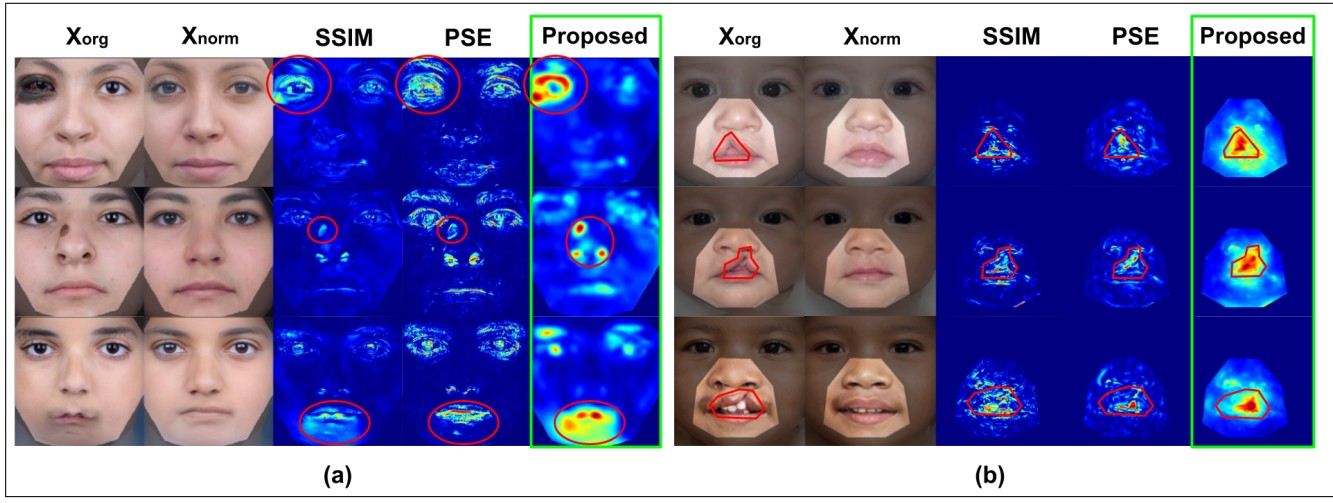

Fig. 3. Effectiveness of using the proposed method to generate anomaly heatmaps for three anomalous faces with different types of cleft and non-cleft facial anomalies ((a) and (b), respectively). Anomalies are more highlighted by the proposed method relative to PSE and SSIM. IRB approvals-written informed consents-were granted to publish (a). Faces in (b) were all fabricated using the CleftGAN generator [12].

TABLE III
$W_1$, MAE, SPEARMAN, AND PEARSON PERFORMANCE MEASURES FOR PATCHCORE AND THE PROPOSED FRAMEWORK WITH SSIM, PSE, LPIPS, AND ALEXNET SIMILARITY MEASURES.

| Method | Performance Metric | | | |
|---|---|---|---|---|
| | $W_1$ | MAE | Spear ($\rho$) | Pearson ($r$) |
| Patchcore [34] | 1.44 | 1.618 | 81.4 | 79.2 |
| SSIM | 0.287 | 0.921 | 81.4 | 83.2 |
| LPIPS | 0.223 | 0.701 | 85.0 | 91.3 |
| PSE | 0.542 | 0.930 | 83.9 | 89.9 |
| AlexNet (proposed) | **0.174** | **0.652** | **85.0** | **92.0** |

Fig. 4 visually demonstrates the relationship between the AlexNet model in the proposed method and the human scores for datasets 1 and 2. The machine scores are highly correlated with human ratings with a Pearson correlation coefficient of 0.92 for cleft lip images and 0.87 for facial images with general abnormalities. Several heatmaps generated by different methods for the oral/nasal region of some fabricated cleft faces are displayed in Fig. 3b.

The proposed anomaly detector allows for the classification of the extracted noise according to the size of the noise blob by taking into account the differential map corresponding to a specific layer of the proposed method. For example, using the difference map from the first layer helps detect the smallest anomalies, as in Fig. 5.

*A. Discussion*

**What types of facial anomalies can be detected by the proposed framework?** Congenital or acquired facial deformities such as cleft lip, pigmented skin lesions, scars, syndromic craniofacial distortions, facial palsy, etc., are examples of semantic anomalies that can be detected by the proposed framework. Additional noise components may be present in

both the $x_{org}$ as well as $x_{norm}$. However, in the process of detecting semantic anomalies, the proposed model cancels them out since they are present in both images (see Section III-C). Another source of irrelevant information that may be filtered out by the proposed model is the normal facial features generated by the imperfect reconstruction of the face (Section III-B). The overall framework aims at detecting different types of structural anomalies rather than noise that will not semantically alter the facial image if present. In the proposed method, fewer processing stages are conducted. This helps the method to be more robust to noise by avoiding extra unnecessary post-processing steps that require additional tuning, as opposed to [13]. Additionally, the suggested approach avoids the need to train a weighting model for each extracted difference map during the comparison phase (see Section III-C). This feature is noteworthy because LPIPS, which is the current standard method of image comparison, learns a weighting model to optimize similarity detection performance on its training dataset. That may not be the most effective model for computing the similarity between anomalous human faces and their normalized counterparts, according to our study results. This indicates that general feature extractors outperform weighted feature extractors in the domain of human facial images. The proposed framework may fail if it does not receive high-quality preprocessed facial images. Another issue is that anomalous crying baby faces may be normalized to smiling rather than normal crying faces. Additionally, lighting conditions and the clarity of the anomalous regions in the face can impact performance. Therefore, the rating framework relies heavily on the actual features present in $x_{org}$.

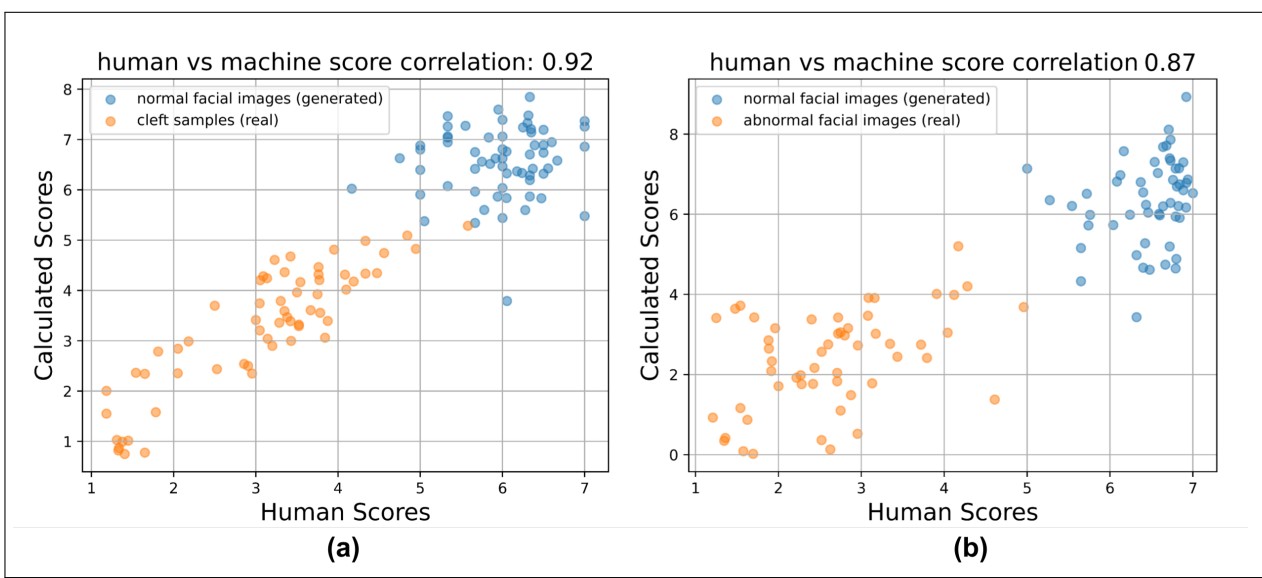

Fig. 4. Correlation between human and machine scores using the Notch Discriminator for (a) the 65 cleft faces and 55 normal faces under analysis (Dataset 1) and (b) the 55 non-cleft abnormal faces and 55 normal faces under analysis (Dataset 2).

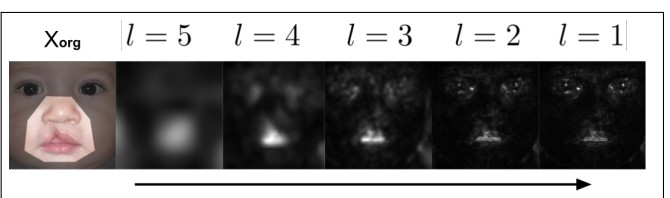

Fig. 5. Anomalies at diverse scales are captured by considering the difference maps associated with different convolutional layers. Early layers detect small-scale noise, while late layers highlight the large-scale anomaly details. The presented face was generated using the CleftGAN generator [12].

## VI. Conclusions

In this study, we investigated the ability of convolutional neural networks to detect abnormal differences between images with facial deformities and their normalized analogues. We showed that unweighted deep feature maps extracted from CNNs are effective in detecting anomalies in facial images. We tested the proposed framework using various types of CNNs and found that the general extracted feature maps from CNNs consistently achieved higher accuracy in detecting the actual abnormal differences, without being necessary to specialize their use to specific anomaly detection datasets and without requiring extra post-processing steps. Furthermore, the scores obtained from the CNN-generated heatmaps were consistent with human ratings of the facial images. The proposed framework functions independently of color information within images, and our computer simulations demonstrate its superior performance relative to state-of-the-art deep CNN-based similarity measures with learned weights such as LPIPS, and shallow image comparison tools such as SSIM and PSNR.

## Acknowledgment

This publication was made possible by grant QNRF-NPRP 13S-0119-200108. The statements made herein are solely the responsibility of the authors.

*Index Terms*—Anomaly Detection, Generative Adversarial Networks, Deep Feature Comparison

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
