# OpenReview forum: "Unsupervised Deep Comparison For Face Anomaly Appraisal"
_IEEE.org/EMBS/BHI/2024/Conference — IEEE BHI'24_

### Official Review · Reviewer_J8nN · 2024-08-05
**Excellent approach !**

**Overall Rating:** 7
**Confidence:** 4

**Other Quality Metrics:**

(a) Clarity of writing; Excellent
(b) Clinical Significance; Great
(c) Methodological Novelty; Good
(d) Experiments and Results: Fair

**Questions For The Authors:**

Could you elaborate on the types of facial anomalies that might present challenges for your framework? Specifically, are there cases where the normalization process or the deep feature comparison might fail to detect subtle but clinically significant anomalies?

Have you tested your framework on a broader range of facial anomalies beyond cleft lip deformities? If so, how does the framework perform on different types of anomalies, and what adjustments, if any, were necessary?

**Strengths:**

The idea is simple, but very understandable and works very well. I wish I came myself with such idea.

One of the significant strengths of the paper is its simplicity and effectiveness. The method is intuitive, as it relies on the normalization of facial images and the comparison of deep features to detect anomalies. This approach is not only easy to understand but also aligns closely with how humans perceive facial abnormalities. The use of the StyleGAN2 generator for image normalization is particularly noteworthy, as it allows for a consistent baseline against which anomalies can be measured.

Moreover, the framework demonstrates robust performance across different datasets, showing high correlation with human assessments (0.92 Pearson’s r score). This high correlation underscores the method's potential to serve as a reliable tool in clinical settings, where objective and consistent evaluation of facial anomalies is crucial.

**Summary Of The Paper:**

The authors use style to gan to "fix" a face making it statistically normal, and compare it with the original face, identifying abnormalities.

The authors introduce a framework that leverages the StyleGAN2 facial generator to normalize human faces, which may exhibit various deformities, and then compares the original and normalized faces using deep features extracted from a pre-trained Convolutional Neural Network (CNN). The difference between these images is used to generate anomaly maps, which are then utilized to produce heatmaps highlighting abnormal regions of the face. The degree of anomaly is quantified by a machine score, which correlates highly with human judgment, making the framework a promising tool for clinical anomaly detection and appraisal.

**Weaknesses:**

Results in other datasets and maybe comparison with other methods would have been nice to see.

While the paper is strong in many aspects, there are areas where it could be further strengthened. The authors could have included more extensive comparisons with other state-of-the-art methods. While they do compare their approach with existing similarity measures such as LPIPS, SSIM, and pixelwise subtraction error (PSE), additional comparisons with other anomaly detection frameworks or methods used in clinical practice could have provided a more comprehensive evaluation of their approach.

Furthermore, the paper would benefit from exploring the limitations of the proposed method in greater detail. For instance, discussing potential borderline cases where the method might struggle or fail to detect anomalies would provide a more balanced view of the framework's applicability.

---

### Official Review · Reviewer_mQyu · 2024-08-15
**Generative model provides metrics of facial anomaly that correlate with human ratings**

**Overall Rating:** 7
**Confidence:** 4

**Other Quality Metrics:**

(a) Clarity of writing; Excellent
(b) Clinical Significance; Great
(c) Methodological Novelty; Good
(d) Experiments and Results; Great

**Questions For The Authors:**

Tables 2 and 3 should clearly indicate which one is the proposed model.  I assume it is “Alex” since that is the model with the best results with bold fonts, but it is not clear. The writeup for Table 2 is also a bit confusing. One is the LPIPS implementation (which is prior work) and the other is the original AlexNet CNN.  Which one is the proposed model that the next sentence refers to?
I assume that the IRB # was not included due to the double-blind nature of the review process.  But I want to make sure the actual IRB study will be included if the paper is accepted, given the sensitive nature of those images.

**Strengths:**

This is an interesting and application of generative AI models to a medical problem.  Given the ongoing debate about the ethics of AI, and in particular generative AI, this is a welcome application that highlights the societal benefits of the latter.
The paper is well written and organized, presents a compelling problem and relates it well with related work in the area.  The paper is accessible to a broad audience but has sufficient technical details for specialists in the area. The methodology is sound and well justified.
Results are significant and laid out well.  The comparison against human ratings is welcome and support the significance of the proposed model. Overall, this is an excellent paper on the applied side of AI techniques on the health domain, and therefore a great fit for BHI.

**Summary Of The Paper:**

This paper presents a generative model to analyze anomalies in human faces.  The approach consists of two main building blocks: a CNN feature extractor that generates a feature map for a given face that has visual anomalies (the “anomaly” face), and a normalization algorithm that generates a “normal” face that best resembles the source face but without the anomalies.  A final anomaly map is generated by comparing the feature maps for the “anomaly” and “normal” faces.  An anomaly metric is then generated from the anomaly map and compared against human ratings of facial anomaly.  The proposed metric shows a strong correlation with human ratings that exceeds that of existing methods.

**Weaknesses:**

While the introduction and related work are excellent, it has a bit too many twists.  I would recommend reorganizing these sections a little bit to make a simpler “story arch” that starts with the background, then a specific problem, then a specific solution.
The proposed model is based on existing GAN and CNN architectures, so its novelty from the perspective of deep learning is not great.  But considering that this is an application study and the scope of the BHI conference, I am not that concerned about it.

---

### Decision · Program_Chairs · 2024-09-23

Accept